# Estimation of Productivity in Dryland Mediterranean Pastures: Long-Term Field Tests to Calibration and Validation of the Grassmaster II Probe

**João Serrano** [1,*], **Shakib Shahidian** [1], **Francisco Moral** [2], **Fernando Carvajal-Ramirez** [3] and **José Marques da Silva** [1,4]

[1]    MED-Mediterranean Institute for Agriculture, Environment and Development, Instituto de Investigação e Formação Avançada, Universidade de Évora, Pólo da Mitra, Ap. 94, 7006-554 Évora, Portugal; shakib@uevora.pt (S.S.); jmsilva@uevora.pt (J.M.d.S.)

[2]    Departamento de Expresión Gráfica, Escuela de Ingenierías Industriales, Universidad de Extremadura, Avenida de Elvas s/n, 06006 Badajoz, Spain; fjmoral@unex.es

[3]    Department of Engineering, Mediterranean Research Center of Economics and Sustainable Development (CIMEDES), University of Almería (Agrifood Campus of International Excellence, ceiA3), La Cañada de San Urbano, s/n. 04120 Almería, Spain; carvajal@ual.es

[4]    AgroInsider Lda. (spin-off da Universidade de Évora), PITE, R. Circular Norte, NERE, Sala 18, 7005-841 Évora, Portugal

[*]    Correspondence: jmrs@uevora.pt; Tel.: +351-266-760-800

**Abstract:** The estimation of pasture productivity is of great interest for the management of animal grazing. The standard method of assessing pasture mass requires great effort and expense to collect enough samples to accurately represent a pasture. This work presents the results of a long-term study to calibrate a Grassmaster II capacitance probe to estimate pasture productivity in two phases: (i) the calibration phase (2007–2018), which included measurements in 1411 sampling points in three parcels; and (ii) the validation phase (2019), which included measurements in 216 sampling points in eight parcels. A regression analysis was performed between the capacitance (CMR) measured by the probe and values of pasture green matter and dry matter (respectively, GM and DM, in kg ha$^{-1}$). The results showed significant correlations between GM and CMR and between DM and CMR, especially in the early stages of pasture growth cycle. The analysis of the data grouped by classes of pasture moisture content (PMC) shows higher correlation coefficients for PMC content >80% (r = 0.775; p < 0.01; RMSE = 4806 kg ha$^{-1}$ and CV$_{RMSE}$ = 28.1% for GM; r = 0.750; p < 0.01; RMSE = 763 kg ha$^{-1}$ and CV$_{RMSE}$ = 29.7% for DM), with a clear tendency for the accuracy to decrease when the pasture vegetative cycle advances and, consequently, the PMC decreases. The validation of calibration equations when PMC > 80% showed a good approximation between GM or DM measured and GM or DM predicted (r = 0.959; p < 0.01; RMSE = 3191 kg ha$^{-1}$; CV$_{RMSE}$ = 23.6% for GM; r = 0.953; p <0.01; RMSE = 647 kg ha$^{-1}$ and CV$_{RMSE}$ = 27.3% for DM). It can be concluded that (i) the capacitance probe is an expedient tool that can enable the farm manager to estimate pasture productivity with acceptable accuracy and support the decision-making process in the management of dryland pastures; (ii) the more favorable period for the use of this probe in dryland pastures in a Mediterranean climate, such as the Portuguese Alentejo, coincides with the end of winter and beginning of spring (February–March), corresponding to PMC > 80%.

**Keywords:** capacitance probe; pasture productivity; dryland pastures; prediction models

## 1. Introduction

The ability to monitor and map pasture biomass in extensive grazing systems provides farmers with vital information for making timely livestock management decisions, such as daily pasture allocation, set stocking rates or rotation interval through the various fields [1]. Estimation of pasture productivity at various stages of the growth cycle is an important element for planning and to calculate forage needs and supplementary feeding [2], an essential pathway to increasing the efficiency of grazing systems [1].

Measurement of pasture productivity can be carried out through direct or indirect methods. The conventional (direct) method is based on harvesting biomass at specific sampling areas, which is a lengthy and expensive process given the large number of samples necessary to accurately represent a field [3], and farmers cannot make this effort in day-to-day management [4]. Since this methodology is not practical at the farm level, other indirect techniques have emerged that provide an estimate of productivity and its spatial variability in large areas in a timely manner, fulfilling one of the important prerequisites of implementing innovative Precision Agriculture strategies [5,6]. With recent advancements in information technologies, remote and proximal sensing and geospatial analyses supported by global positioning systems, it is increasingly possible to identify and analyze the temporal and spatial variability within fields, to maximize the yield and protect the environment [7].

One of the methods proposed for quantifying and mapping the pasture production variability is based on the measurement of spectral vegetation indices, mainly the NDVI (normalized difference vegetation index and Equation (1), calculated by measuring the reflectance of the radiation emitted by the plants at certain wavelengths, using satellite images [1,8–10]. This information can be collected by satellite remote sensing imagery (RS), unmanned aerial vehicles (UAV) or by means of ground-based vehicle mounted proximal sensors [7].

$$\text{NDVI} = \frac{\text{NIR} - \text{Red}}{\text{NIR} + \text{Red}} \tag{1}$$

where NIR is near infrared radiation; and Red is visible red radiation.

Although the use of satellite imagery is a very promising, low-cost and non-destructive technique, it has its own limitations. The restrictions known for applications of remote sensing systems in farm management include the following: (a) the gathering and delivery of images in an exceedingly timely manner; (b) the shortage of high spatial resolution, image interpretation and data extraction issues; and (c) the combination of those data with agronomic knowledge into expert systems [7]. Handcock et al. [11] highlight the difficulties of RS resulting from of spatial resolution and the presence of clouds, as well as the spatial and temporal specificity of the associated algorithms. The limitation imposed by the presence of clouds can be overcome by using UAV equipped with the appropriate sensors [7]. In either case, whether using satellites or UAV, an important limitation persists in the case of *montado* agro-silvo-pastoral system: it is not possible to access information about the pasture under tree canopy, and this has led to the use of proximal sensors [11]. In the last few years, many such sensors have been developed and marketed, such as Crop Circle (Holland Scientific, Lincoln, NE, USA & AgLeader, Ames, IA, USA), Yara N-Sensor (Yara, Oslo, Norway & Agricon, Ostrau, Germany), GreenSeeker (N-Tech, Ukiah, CA, USA & Trimble, Sunnyvale, CA, USA) and OptRx (AgLeader, Ames, IA, USA) [7,12]. Despite the growing use of optical sensors for monitoring the vegetation cover of the globe (either through remote or proximal detection), NDVI tends to saturate for high values of plant leaf area indices, which is when productivity has reached high values [1,9]. Figure 1 shows the evolution pattern over the year of NDVI, measured by an active optical sensor (OptRx), of green matter (GM) and dry matter (DM) of a pasture in Mitra farm, Alentejo, mean of three years, 2015/2016, 2016/2017 and 2017/2018, based on work published by Serrano et al. [2]. It is evident that there is a drop in NDVI from February to March, while GM still increases in April and DM increases until June. This behavior confirms the limitations resulting from the use of NDVI to estimate pasture productivity.

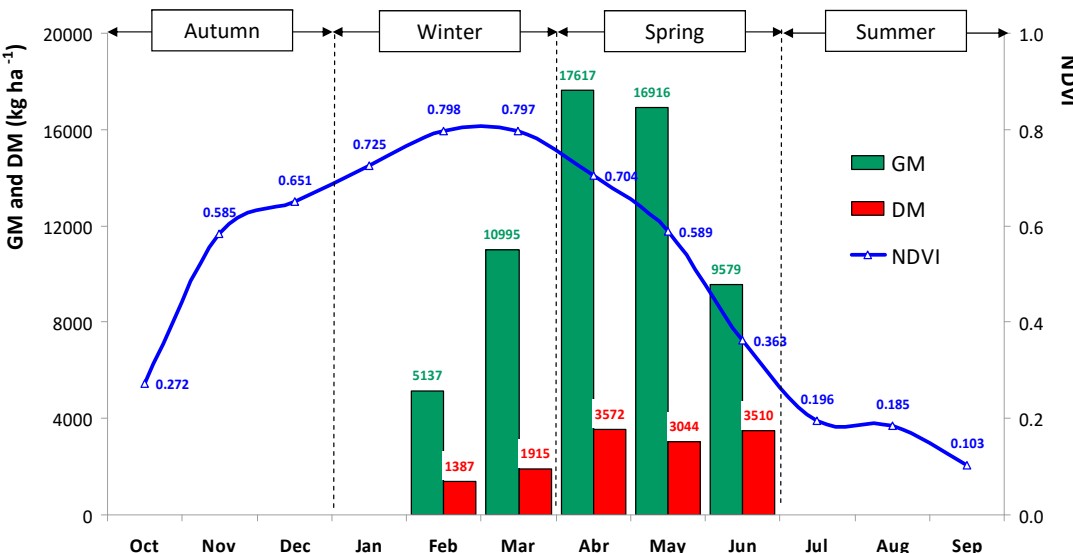

**Figure 1.** Evolution pattern over the year of normalized difference vegetation index (NDVI), of green matter (GM) and dry matter (DM) of a pasture in Mitra farm, Alentejo, mean of three years, 2015/2016, 2016/2017 and 2017/2018, based on work published by Serrano et al. [2].

Another specific proposal for estimating pasture productivity has been introduced by Novel Ways Electronic, a New Zealand company that has developed a capacitance probe (Grassmaster II; Figure 2), which was described by Serrano et al. [5,12]. This probe is based on the pioneering work of Vickery and Nicol [13], who described the theory and the operating principles of such equipment. Capacitance instruments are generally composed of an electronic circuit, which generates a signal of a certain frequency and then carries out a measurement of the capacitance of the air–herbage mixture. The probe produces an electric field that is modified by the pasture close to it. The modified field is detected as a change in capacitance by the electronic circuit within the probe. This capacitance change is proportional to the water content of the grass. Because the dry matter content is highly correlated with the water content, the probe capacitance change is calibrated to the pasture yield, with linear calibration equations used to correlate readings taken on a pasture over a short seasonal time, with measured dry matter production [4,5]. The manufacturer has presented an equation with which the computer module automatically calculates pastures' productivity based on the readings from the probe (Equation (2)). Nonetheless, this equation was calculated for New Zealand pastures, which are composed of legumes and grasses, with an average dry matter of 14–16%.

$$\mathrm{DM} = 0.72 \times \mathrm{CMR} - 2200 \tag{2}$$

where DM is the estimated productivity of the pasture, in kg of dry matter per hectare; and CMR is the capacitance measured by the Grassmaster II probe.

The relation between pasture productivity and capacitance measured by the probe is influenced by factors such as the mixture of plant species and their phenological stage [4,12,14,15]. Seasonal adjustments to calibration equations are necessary, as the moisture content of pasture vegetation varies with phenological stages and live/dead material ratio [12,16]. This variability requires the calibration and validation tests to span the diversity of parameters related to permanent dryland pastures of Alentejo area of Portugal. The growth and development of these pastures is conditioned by a series of factors, such as soil fertility, the grazing strategy and, mainly, by the rainfall pattern, which introduces an interannual variation in terms of both growth and development. Thus, it is difficult to conceive that a generic equation can adequately portray the seasonal variability associated with the Mediterranean climate, which is accentuated by the complexity of the *montado* ecosystem [6]. In this context, this work presents the results of tests carried out in a long-term study between 2007 and 2019 to calibrate

and validate a capacitance probe (Grassmaster II) to estimate pasture productivity in the *montado* Mediterranean ecosystem and to determine the most suitable time to use this probe.

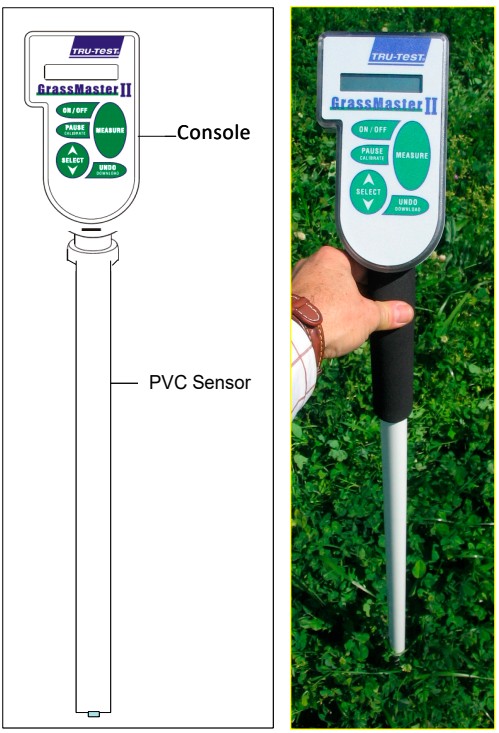

**Figure 2.** Grassmaster II: electronic interface (console) and probe (PVC Sensor).

## 2. Material and Methods

### 2.1. Characteristics of the Experimental Sites

The experiments were carried out in two phases (Figure 3): (i) calibration phase, between 2007 and 2018, included measurements at three fields located in the Évora District (Figure 3); (ii) validation phase, between February and June 2019, included measurements at eight parcels located predominantly in Alentejo region (Figure 4). The characteristics of these eleven sites are presented in Table 1.

These annual or permanent biodiverse pastures (composed of different botanical species: legumes, grasses, composites and other species) are representative of the regional dryland pastures, with common characteristic in terms of a Holm oak or Cork oak *montado*, grazed by sheep or cattle in a rotational or permanent system. The fact that they are located on soils that are poor, relatively acidic and deficient in phosphorous leads some farmers to try to improve the productivity through application of dolomitic limestone amendments and phosphorous fertilizers [11].

### 2.2. Pasture Sampling

Pasture sampling in the calibration phase was carried out at each location, with repetitions through the growth cycle (between February and June), totaling 1411 samples, representative of the three fields: "Mitra" (4.3 ha; 726 samples, taken during six years: 2007, 2013 and between 2015 and 2018); "Revilheira" (6.1 ha; 199 samples, taken during two years, 2007 and 2013) and "Silveira" (7.2 ha; 486 samples, taken during four years, between 2013 and 2016). Pasture sampling in the validation phase was carried out between February and June 2019, in eight experimental fields with area of approximately 25 ha ("Azinhal", "Cubillos", "Grous", "Mitra B", "Murteiras", "Padres", "Quinta França" and "Tapada"). In each of these eight fields, nine samples that were representative of the pasture spatial variability were taken at three different times, for a total of 216 samples (8 fields × 9 samples × 3 times).

The sampling process consisted of measuring pasture capacitance with the Grassmaster II probe, followed by placing a metal quadrat with an area of 0.1 m$^2$ (0.25 m × 0.40 m in size) over the pasture. The pasture contained in this area was cut with electric shears and preserved in numbered plastic bags. Once in the laboratory, the pasture sample was weighed to establish total biomass produced by unit area (GM in kg ha$^{-1}$), dried in an oven (72 h at 65 °C) and weighed again to establish productivity in terms of dry matter per unit of area (DM in kg ha$^{-1}$) and PMC (pasture moisture content or green matter moisture content wet basis, in %).

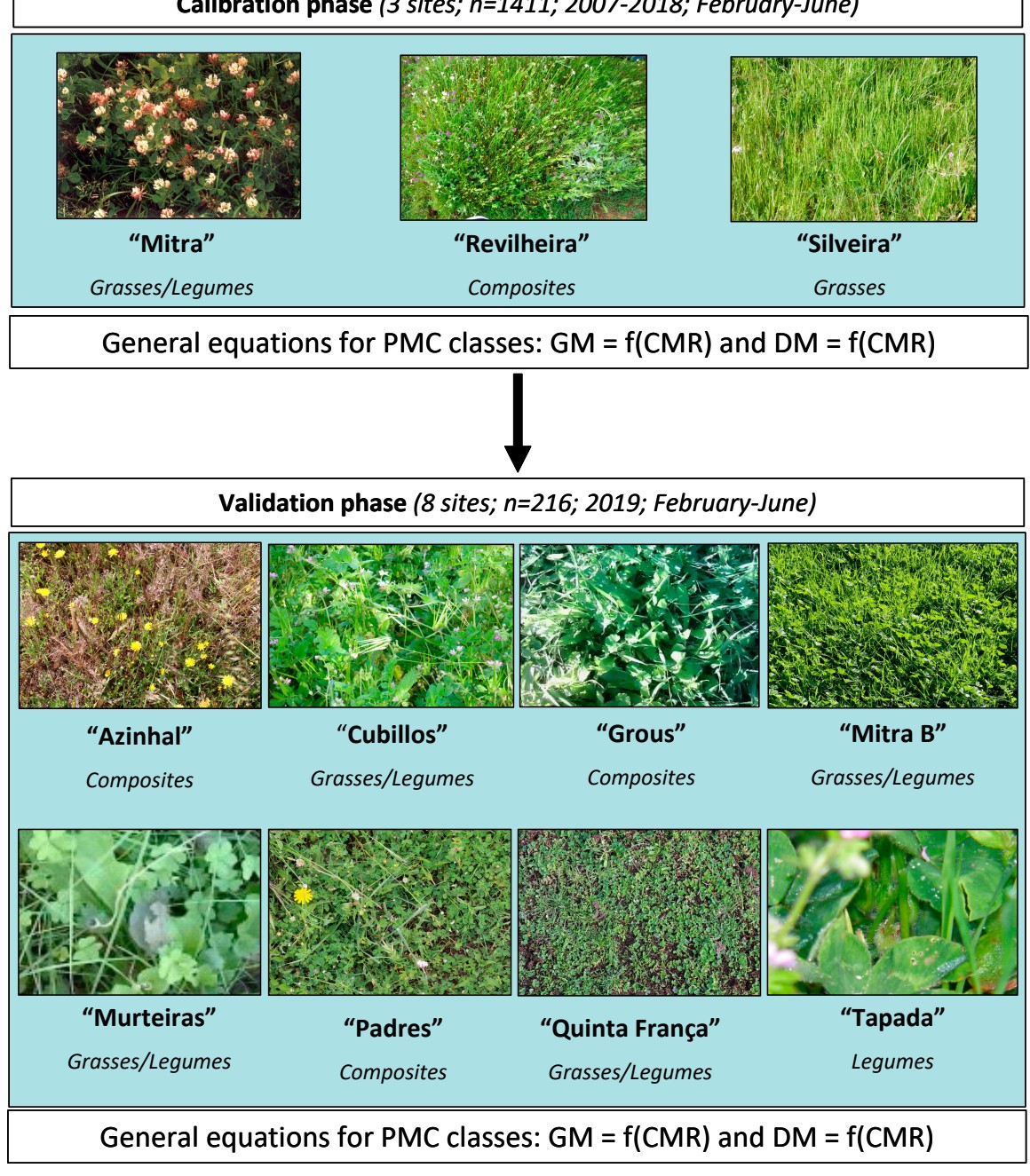

**Figure 3.** Schematic representation of the experimental methodology used in this work.

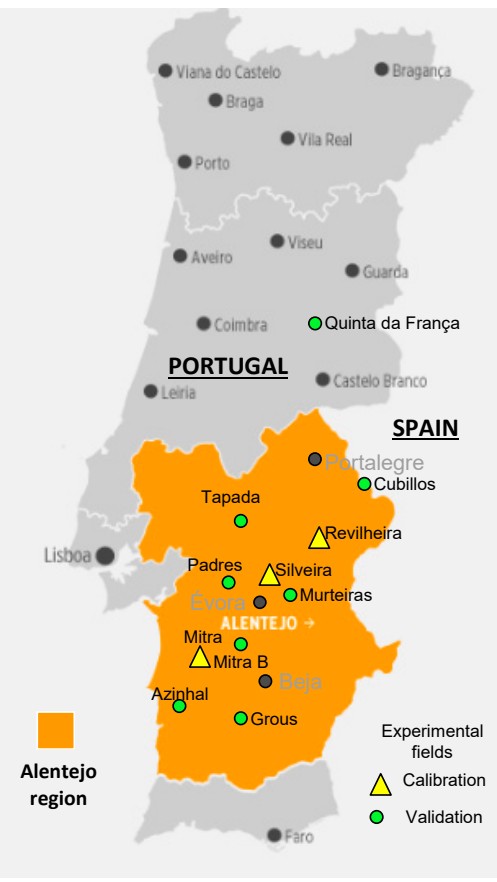

**Figure 4.** Location of experimental farms in Alentejo region of Southern Portugal.

**Table 1.** Characteristics of the eleven experimental fields used in this study.

| Site (Number of Samples) | Coordinates | Pasture Type | Predominant Trees | Animal Species | Sampling Years |
|---|---|---|---|---|---|
| "Mitra" (n = 726) | 38°32.2′ N; 8°01.1′ W | Permanent; biodiverse (mixture of grasses and legumes) | Holm oak | Sheep in permanent grazing | 2007; 2013; 2015–2018 |
| "Revilheira" (n = 199) | 38°27.9′ N; 7°25.7′ W | Permanent; biodiverse (predominance of composites) | Holm oak and Cork oak | Sheep in rotational grazing | 2007; 2013 |
| "Silveira" (n = 486) | 38° 62.2′ N; 7°94.8′ W | Permanent; biodiverse (predominance of grasses) | Olive, Holm oak and Mulberries | Sheep in permanent grazing | 2013–2016 |
| "Azinhal" (n = 27) | 38°6.2′ N; 8°26.9′ W | Permanent; biodiverse (predominance of composites) | Holm oak and Cork oak | Sheep in rotational grazing | 2019 |
| "Cubillos" (n = 27) | 39°10.0′ N; 6°44.6′ W | Annual; biodiverse (mixture of grasses and legumes) | Holm oak and Cork oak | Cattle in rotational grazing | 2019 |
| "Grous" (n = 27) | 37°52.3′ N; 7°56.7′ W | Permanent; biodiverse (predominance of composites) | Holm oak | Cattle in rotational grazing | 2019 |
| "Mitra B" (n = 27) | 38°31.8′ N; 8°0.9′ W | Permanent; biodiverse (mixture of grasses and legumes) | Holm oak and Cork oak | Cattle in rotational grazing | 2019 |
| "Murteiras" (n = 27) | 38°23.4′ N; 7°52.5′ W | Annual; biodiverse (mixture of grasses and legumes) | Holm oak and Cork oak | Sheep in permanent grazing | 2019 |
| "Padres" (n = 27) | 38°36.4′ N; 8°8.7′ W | Permanent; biodiverse (predominance of composites) | Holm oak | Cattle in permanent grazing | 2019 |
| "Quinta França" (n = 27) | 40°16.4′ N; 7°25.9′ W | Permanent; biodiverse (mixture of grasses and legumes) | Eucalyptus | Sheep and cattle in rotational grazing | 2019 |
| "Tapada" (n = 27) | 39°9.5′ N; 7°31.9′ W | Permanent; biodiverse (mixture of legumes) | Holm oak and Cork oak | Cattle, sheep or pigs in rotational grazing | 2019 |

### 2.3. Statistical Analysis

The statistical analysis of the results included a descriptive analysis with calculation of average and standard deviation (SD) of each dataset. An analysis of the linear correlations between capacitance (CMR) measured by the probe and values of pasture productivity (pasture green matter and pasture dry matter, respectively, GM and DM, in kg ha$^{-1}$) was carried out with the MSTAT-C software, with a significance level of 95% ($p < 0.05$). The data were organized and analyzed by pasture moisture content (PMC) classes. The Pearson (r) correlation coefficient was used to measure the degree of correlation or the linear dependence between variables and the coefficient of determination ($R^2$), to measure the proportion of the total variation of the dependent variable explained by the variation of the independent variable. The rigor of the resulting regression models was evaluated by the absolute value of the root mean square error (RMSE; Equation (3)) and its relative value (CV$_{RMSE}$; Equation (4)). This statistical parameter measures the average magnitude of the error resulting from the estimate.

$$\text{RMSE} = \sqrt{\frac{\sum_{i=1}^{n} (E_i - M_i)^2}{n}} \tag{3}$$

$$\text{CV}_{RMSE} = \frac{\text{RMSE}}{\bar{\bar{y}}} \times 100 \tag{4}$$

where n is the number of observations; $E_i$ and $M_i$ are the estimated and observed (measured) values, respectively; and $\bar{y}$ is the average value of each measured parameter.

## 3. Results and Discussions

### 3.1. Variability Pattern of the Measured Parameters

Tables 2 and 3 present the average values and the standard deviation of the parameters measured in the monitored pastures, in each PMC class considered, in calibration and validation phases, respectively.

**Table 2.** Mean ± standard deviation of the parameters measured in the pastures of the three fields used in the calibration phase.

| PMC Classes, % | n | CMR | GM, kg ha$^{-1}$ | DM, kg ha$^{-1}$ |
|:---:|:---:|:---:|:---:|:---:|
| >85 | 236 | 8772 ± 2887 | 23,696 ± 17,570 | 2923 ± 2067 |
| 80–85 | 331 | 7523 ± 2370 | 14,351 ± 9374 | 2444 ± 1564 |
| 75–80 | 231 | 6488 ± 2354 | 11,587 ± 8753 | 2561 ± 1930 |
| 70–75 | 187 | 6428 ± 2429 | 9950 ± 6712 | 2729 ± 1834 |
| 65–70 | 143 | 5727 ± 2624 | 8645 ± 9825 | 2775 ± 1756 |
| 60–65 | 118 | 5645 ± 1726 | 8389 ± 5892 | 3119 ± 2202 |
| <60 | 165 | 4586 ± 1143 | 7023 ± 5089 | 4586 ± 1143 |
| >80 | 567 | 8043 ± 2668 | 17,119 ± 11,341 | 2570 ± 1657 |
| 70–80 | 418 | 6461 ± 2385 | 10,854 ± 7939 | 2636 ± 1887 |
| <70 | 426 | 5262 ± 1631 | 7946 ± 5509 | 3175 ± 2313 |

PMC—pasture moisture content; n—number of samples; CMR—Grassmaster II measurements; GM—green matter; DM—dry matter.

**Table 3.** Mean ± standard deviation of the parameters measured in the pastures of the eight fields used in the validation phase.

| PMC, % | n | CMR | GM, kg ha$^{-1}$ | DM, kg ha$^{-1}$ |
|:---:|:---:|:---:|:---:|:---:|
| >80 | 47 | 6651 ± 3130 | 13,516 ± 8974 | 2364 ± 1576 |
| 70–80 | 98 | 6268 ± 3091 | 11,782 ± 7814 | 2901 ± 1737 |
| <70 | 71 | 4962 ± 1231 | 6154 ± 3955 | 2310 ± 1334 |

PMC—pasture moisture content; n—number of samples; CMR—Grassmaster II measurements; GM—green matter; DM—dry matter.

The spatial variability, detected by the high coefficients of variation of the pasture productivity, GM and DM (around 50–80%; Tables 3 and 4) and Grassmaster II measurements (CV of around 30%), indicates the suitability of employing differentiated management techniques, which fits in the Precision Agriculture concept.

**Table 4.** Linear regression equations proposed to estimate pasture green matter yield (GM), based on Grassmaster II measurements (CMR), for each pasture moisture content (PMC) class considered.

| PMC Classes, % | N | Linear Equations | r | RMSE, kg ha$^{-1}$ | CV$_{RMSE}$, % |
|---|---|---|---|---|---|
| >85 | 236 | 3.3319 *CMR − 8225.4 | 0.759 ** | 5855 | 24.7 |
| 80–85 | 331 | 3.0236 *CMR − 8395.7 | 0.765 ** | 4597 | 32.0 |
| 75–80 | 231 | 2.3271 *CMR − 3512.4 | 0.626 * | 6813 | 58.8 |
| 70–75 | 187 | 1.4044 *CMR + 922.48 | 0.508 * | 5766 | 57.8 |
| 65–70 | 143 | 1.4915 *CMR + 103.8 | 0.477 * | 4852 | 56.1 |
| 60–65 | 118 | 1.46 *CMR + 146.37 | 0.428 * | 5303 | 63.2 |
| <60 | 165 | 2.1234 *CMR − 2714.2 | 0.445 * | 4460 | 63.5 |
| >80 | 567 | 3.2963 *CMR − 9392.7 | 0.775 ** | 4806 | 28.1 |
| 70–80 | 418 | 1.9042 *CMR − 1449.2 | 0.572 * | 6839 | 63.0 |
| <70 | 426 | 1.5881 *CMR − 411.24 | 0.470 * | 4837 | 60.9 |

PMC—pasture moisture content; n—number of samples; CMR—Grassmaster II measurements; GM—green matter; DM—dry matter; r—correlation coefficient; RMSE—root mean square error; CV$_{RMSE}$—coefficient of variation of root mean square error; * correlation significant at the 0.05 level; ** correlation significant at the 0.01 level.

The number of measurements performed in each PMC class in calibration phase is presented in Figure 5. More than 40% of measurements were performed with PMC > 80%, i.e., in the early stages of the pasture vegetative cycle, usually in the months of February and March [17]. This is the period when the probe has the highest sensitivity, given its operating principle: Essentially, this device responds to the wet biomass [16], and according to the manufacturer of the probe, water is, by far, the material that has the greatest effect on the probe signal. On the other hand, the probe does not provide any readings when the pasture is dry (during summer) or when the pasture is wet (rain, frost or dew) [17].

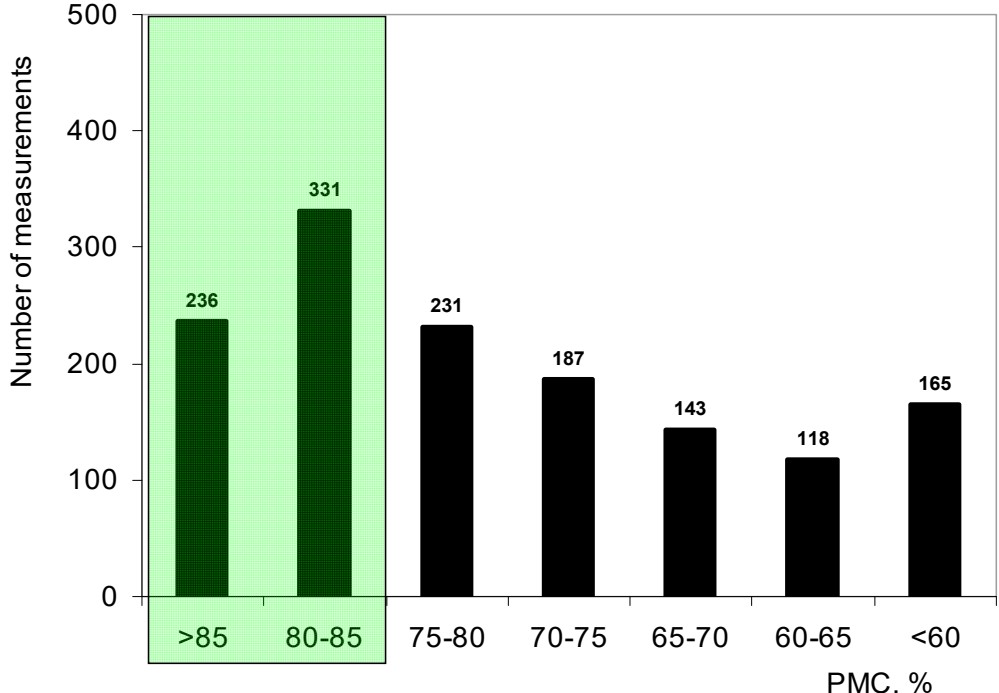

**Figure 5.** Distribution of the number of measurements performed in each pasture moisture content (PMC) class considered in the calibration phase.

### 3.2. Correlations between Pasture Productivity and Grassmaster II Measurements

Tables 4 and 5 show, respectively, the linear regression equations between GM and CMR and between DM and CMR, for each PMC class considered. For each regression equation, the correlation coefficient (r) and the root mean error (RMSE) are presented in absolute value and in percentage ($CV_{RMSE}$).

**Table 5.** Linear regression equations proposed to estimate pasture dry matter yield (DM), based on Grassmaster II measurements (CMR), for each pasture moisture content (PMC) class considered.

| PMC Classes, % | n | Linear Equations | R | RMSE, kg ha$^{-1}$ | $CV_{RMSE}$, % |
|---|---|---|---|---|---|
| >85 | 236 | 0.4592 *CMR − 1281.3 | 0.750 ** | 844 | 28.8 |
| 80–85 | 331 | 0.4983 *CMR − 1304.5 | 0.755 ** | 765 | 31.3 |
| 75–80 | 231 | 0.5065 *CMR − 725.1 | 0.618 * | 1515 | 59.2 |
| 70–75 | 187 | 0.3819 *CMR + 274.19 | 0.506 * | 1577 | 57.8 |
| 65–70 | 143 | 0.4829 *CMR + 9.1169 | 0.479 * | 1562 | 56.3 |
| 60–65 | 118 | 0.5279 *CMR + 138.59 | 0.414 * | 1996 | 64.0 |
| <60 | 165 | 0.7052 *CMR + 327.74 | 0.297 ns | 2581 | 72.5 |
| >80 | 567 | 0.4656 *CMR − 1174.8 | 0.750 ** | 763 | 29.7 |
| 70–80 | 418 | 0.4483 *CMR − 260.57 | 0.567 * | 1543 | 58.5 |
| <70 | 426 | 0.4222 *CMR + 953.26 | 0.298 ns | 2122 | 66.8 |

PMC—pasture moisture content; n—number of samples; CMR—Grassmaster II measurements; GM—green matter; DM—dry matter; r—correlation coefficient; RMSE—root mean square error; $CV_{RMSE}$—coefficient of variation of root mean square error; * correlation significant at the 0.05 level; ** correlation significant at the 0.01 level; ns—not significant.

Figure 6 shows the evolution of the correlation coefficient and coefficient of variation of the root mean square error ($CV_{RMSE}$) of the green matter (GM) and dry matter (DM) prediction equations based on pasture moisture content (PMC) classes considered in the calibration phase. The highest correlation coefficients and the lowest $CV_{RMSE}$ (better predictability) occur for PMC > 80%. As PMC decreases, these indicators deteriorate: the correlation coefficient decreases, and the $CV_{RMSE}$ increases. For this reason, the analysis of the data grouped by PMC classes (>80%; 70–80% and <70%) was also considered (see Tables 2–5) in this study, which is in agreement with suggestions of several authors [18–21]. Figure 7 shows linear regression equations between CMR and pasture productivity (GM and DM) for grouped class of PMC > 80%.

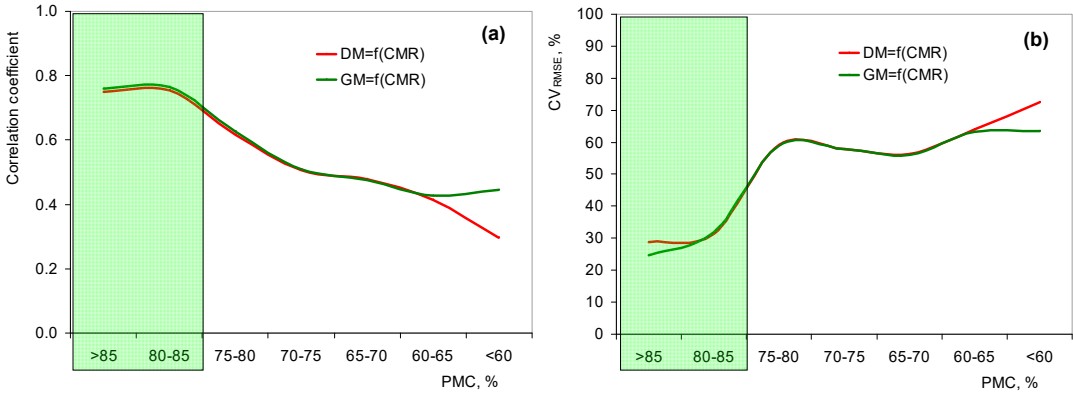

**Figure 6.** Evolution of the correlation coefficient (**a**) and coefficient of variation of the root mean square error ($CV_{RMSE}$; (**b**)) of the green matter (GM) and dry matter (DM) prediction equations, based on pasture moisture content (PMC) classes considered in the calibration phase.

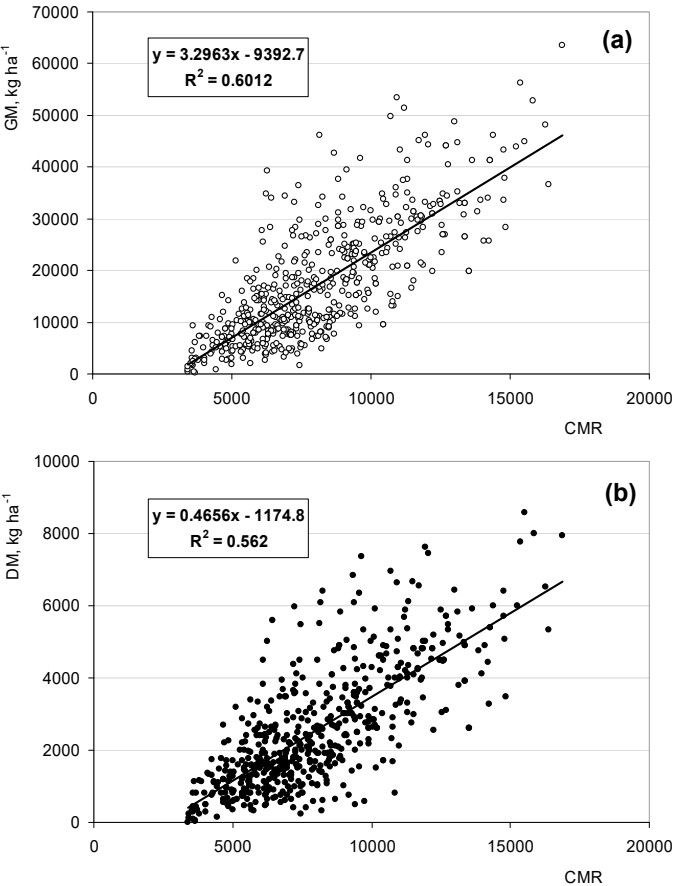

**Figure 7.** Linear regression equations between Grassmaster II measurements (CMR) and pasture productivity for pasture moisture content (PMC) > 80%: (**a**) green matter (GM) versus CMR; (**b**) dry matter (DM) versus CMR.

According to Jamieson et al. [22], this estimate can be considered acceptable ($CV_{RMSE}$ of 28.1% and 29.7%, respectively for GM and DM), especially considering that the application of sensor techniques to evaluate the existing variability is difficult on permanent grassland with diverse species, plant spacing, morphology and color [23]. Furthermore, the situation becomes more complex when grazing animals are involved, which is the case, due to dynamic interactions between plants and animals [24]. This degree of uncertainty associated with calibration and validation of this capacitance probe was also quoted by other authors [3,14,25–27].

Figures 8–10 show the relation between pasture productivity measured in 2019, in eight experimental fields ("Azinhal", "Cubillos", "Grous", "Mitra B", "Murteiras", "Padres", "Quinta França" and "Tapada"—validation phase) and pasture productivity prediction based on the equations obtained in calibration phase for PMC > 80%, 70–80% and <70%, respectively. Figure 8, referring to the PMC class > 80%, confirms the closest approximation between pasture productivity (GM and DM) predicted and measured, proving a very acceptable degree of confidence ($CV_{RMSE}$ of 23.6% and 27.3%, respectively to GM and DM).

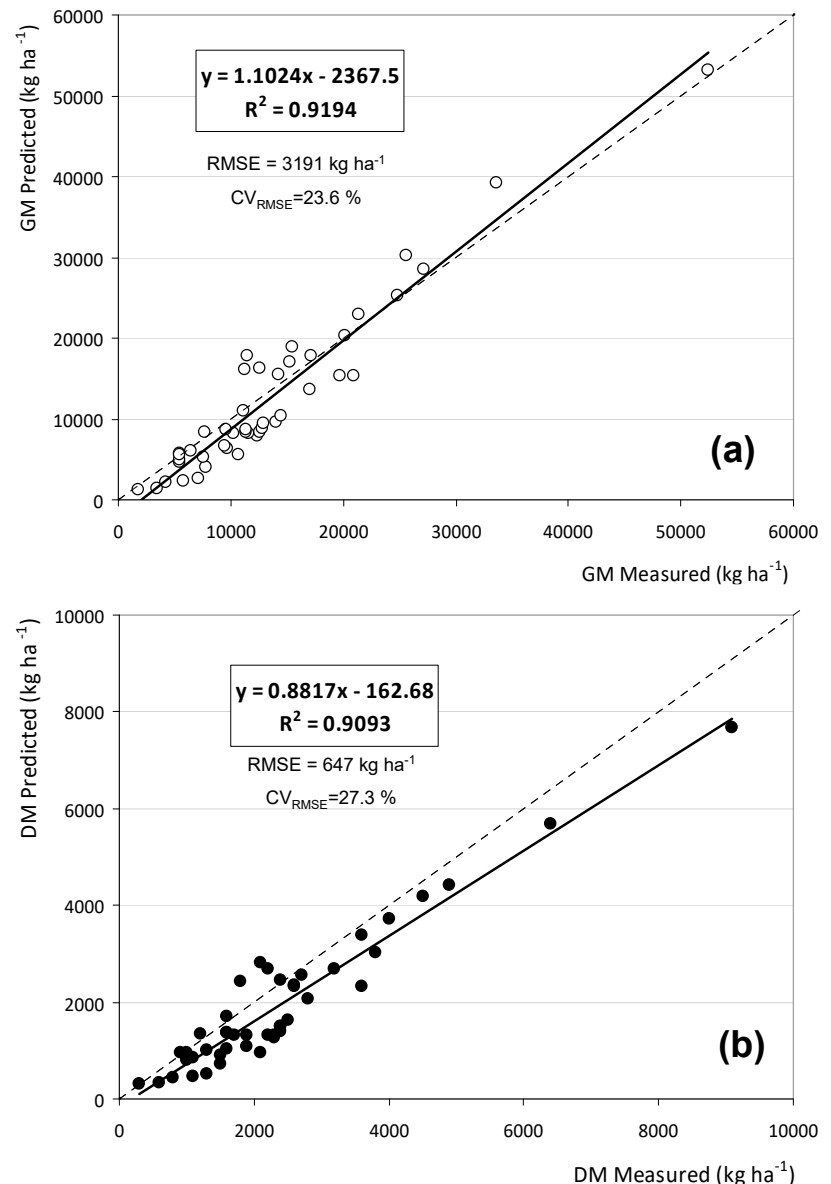

**Figure 8.** Relation between pasture productivity measured ((**a**) green matter; (**b**) dry matter (DM)) in 2019, in eight experimental fields ("Azinhal", "Cubillos", "Grous", "Mitra B", "Murteiras", "Padres", "Quinta França" and "Tapada"—validation phase), and pasture productivity predicted based on the equations obtained in calibration phase for pasture moisture content (PMC) > 80%. $R^2$—coefficient of determination; RMSE—root mean squared error; $CV_{RMSE}$—coefficient of variation of root mean squared error.

These results are very promising, especially given the heterogeneity of biodiverse pastures. Each pasture is a different ecosystem, with specific characteristics, which vary according to the different plant species and their vegetative states [18–20]. Deviations of approximately 25% between the measured and the estimated productivity are, in practice, not impeditive to support precision management decisions in grazing systems, in particular for calculation and dynamic organization of the number of animals per hectare. We believe that better results will only be possible in monospecies pastures, for example, only legumes (such as *Trifolium subterraneum*) or grasses (such as *Lolium multiflorum* Lam.), but very unrepresentative of dryland pastures in the Alentejo region [28].

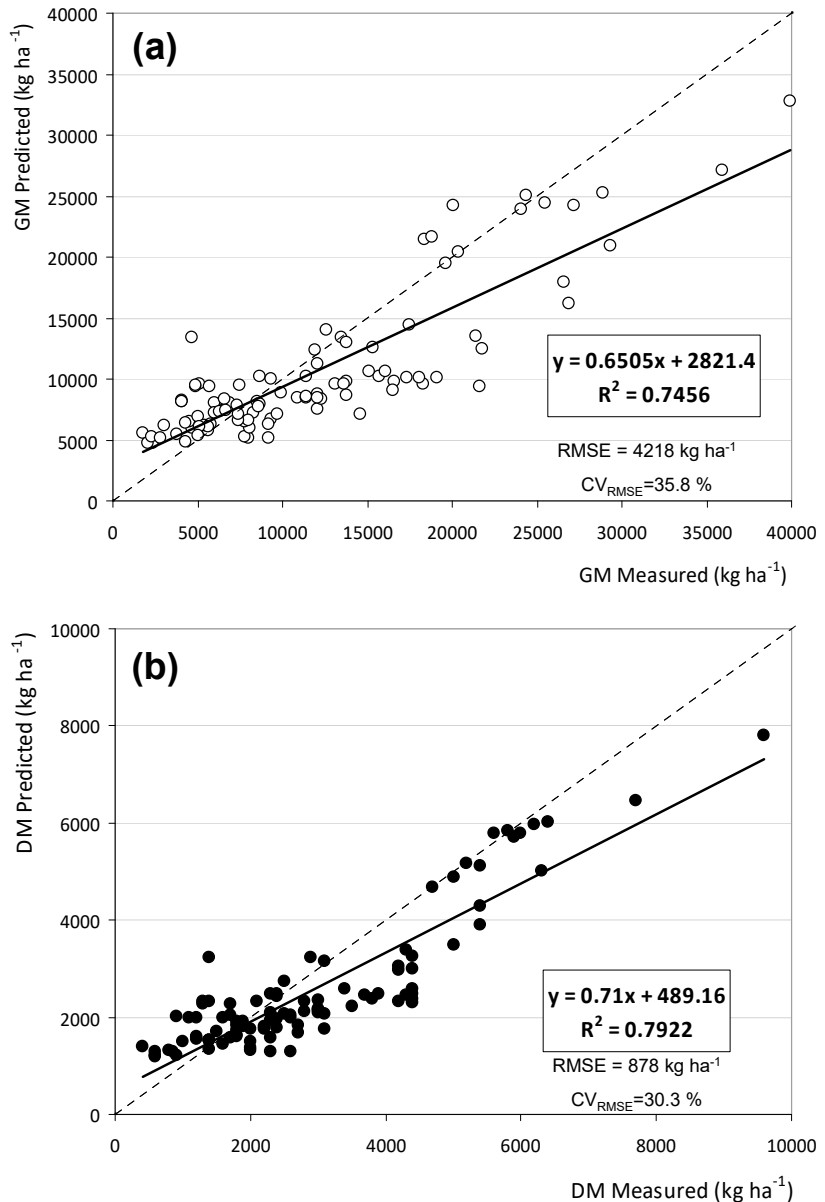

**Figure 9.** Relation between pasture productivity measured ((**a**) green matter; (**b**) dry matter (DM)) in 2019, in eight experimental fields ("Azinhal", "Cubillos", "Grous", "Mitra B", "Murteiras", "Padres", "Quinta França" and "Tapada"—validation phase), and pasture productivity predicted based on the equations obtained in calibration phase for pasture moisture content (PMC) 70–80%. $R^2$—coefficient of determination; RMSE—root mean squared error; $CV_{RMSE}$—coefficient of variation of root mean squared error.

The importance of estimating pasture productivity was already justified in the introduction of this paper. It was also reported that each of the presented sensors or techniques has its own advantages and disadvantages, in the context of high variability conferred by strong spatially variable phenology, morphology, species composition and green vs. dry fraction characteristics of biodiverse pastures [1]. Various techniques have been developed to take indirect measurements of pasture biomass; each of these has strengths and weaknesses in certain situations [10]. Today, agriculture faces challenges related to competitivity and sustainability, which demand from the farm manager an up-to-date knowledge of the existing options for optimizing the productive process [12]. Therefore, it should be noted that electronic capacitance meter Grassmaster II, in the same way as the rising-plate meter or the sward stick for example, fits in the proximal or ground methods that require an operator to carry

out manual point-to-point measurements, and thus cannot provide continuous estimates across large spatially diverse pastoral landscapes without considerable effort [29]. In addition, even the current version (Grassmaster Pro), with an acquisition cost of 1995 NZ $ (about 1100 € plus shipping costs), is somehow limited by its inability to connect with a GNSS (Global Navigation Satellite Systems) receiver, which would have been fundamental from the perspective of mapping spatial variability. These negative aspects put this probe at a disadvantage compared to several other similar pieces of equipment that are available and can potentially connect with GNSS receivers and be mounted on motorized vehicles (agricultural tractors, all-terrain vehicles, etc.), including the optical sensors (referred in the introduction), C-Dax Pasture Meter (Pasture Meter, C-Dax Ltd., Palmerston North, New Zealand) or the Farmworks Ultrasonic Feed Reader (Department of Primary Industries, Orange, Australia) [30]. Figure 11a shows the desirable immediate development for this type of sensors from the perspective of "smart sampling" services assured by agricultural consultancy enterprises: the possibility to install on a mobile platform that can be used for automated measurements and connection to a GNSS antenna.

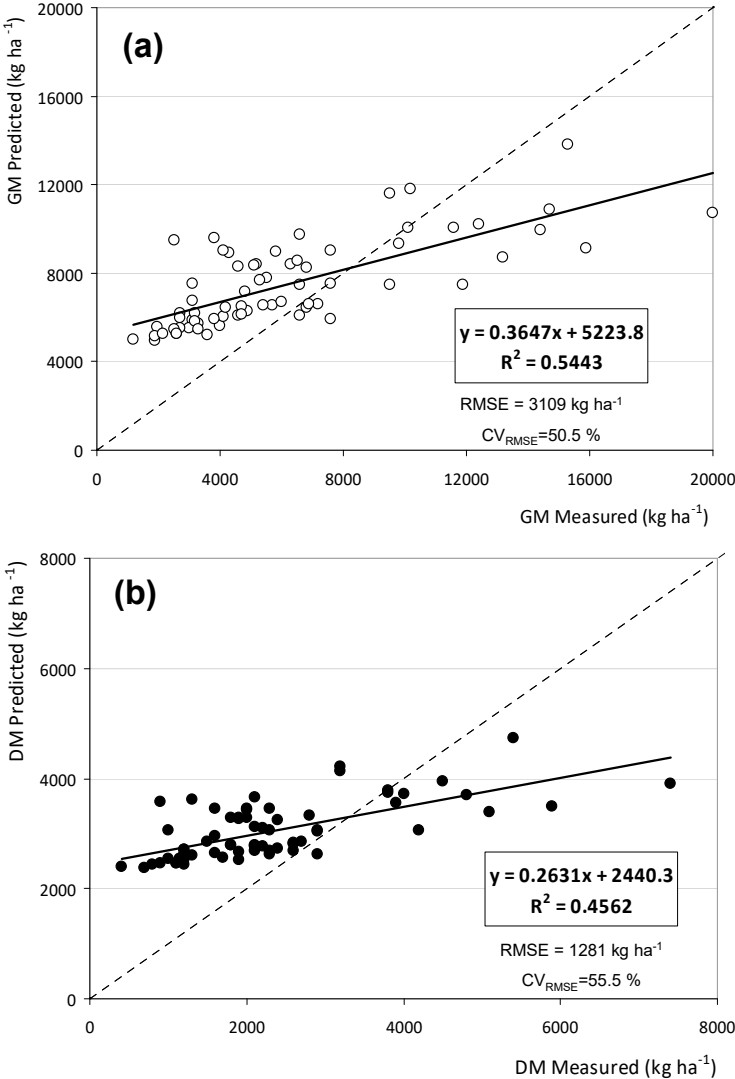

**Figure 10.** Relation between pasture productivity measured ((**a**) green matter; (**b**) dry matter (DM)) in 2019, in eight experimental fields ("Azinhal", "Cubillos", "Grous", "Mitra B", "Murteiras", "Padres", "Quinta França" and "Tapada"—validation phase), and pasture productivity predicted based on the equations obtained in calibration phase for pasture moisture content (PMC) < 70%. $R^2$—coefficient of determination; RMSE—root mean squared error; $CV_{RMSE}$—coefficient of variation of root mean squared error.

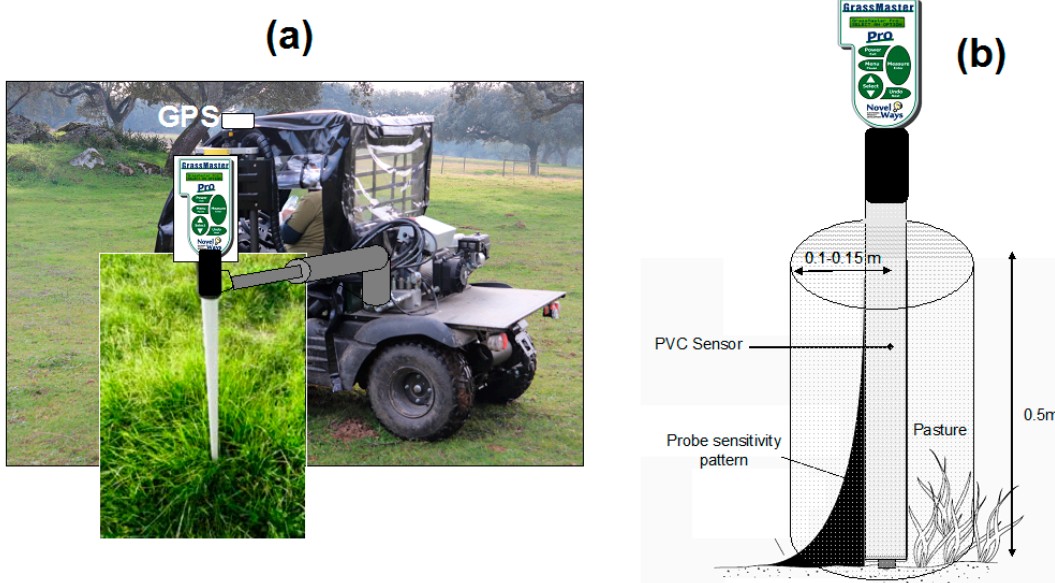

**Figure 11.** (**a**) Desirable development of Grassmaster probe device: portability and georeferencing; (**b**) probe sensitivity pattern.

The operating principle of the Grassmaster II probe, based on sensitivity to PMC [4,5], limits its period of application, with greater accuracy at higher levels of PMC, not allowing, however, precise readings during the final stage of the pasture growth cycle, which may constitute a limitation for the management of animal grazing and food supplementation. This limitation is particularly important because the focus on grazing management is becoming more critical, in an attempt to reduce dependence on expensive imported supplements and to improve farm profitability [31]. Finally, the same operating principle of the Grassmaster II probe (Figure 11b) showed, despite all of this, the potential for accurate biomass estimation in an important time window of the vegetative cycle of dryland pastures in the Mediterranean region. The geometry of the volume scanned by the probe, up to 0.50 m in height and a radius of 0.10–0.15 m, provides this sensor with an important valence, that is, the ability to integrate capacitance measurements of horizontal and vertical structure of pasture, which overcome the limitations of only using pasture height measurements, or of solely using vegetation indices [32]. This can also be an opportunity for improving biomass estimation accuracy from February to March in the Mediterranean region, when pasture productivity reaches high values, leading to the saturation of optical sensors [1]. This probe presents the profile for data integration or data fusion approaches referred to by several authors [1,7,32,33], since it provides information related to physical parameters such PMC and plant height, which can complement and refine the information obtained from spectral reflectance measurements (by remote or proximal sensing) and increase the range over which biomass can be estimated.

## 4. Conclusions

This long-term study (2007–2019) presents and validates equations for estimating pasture productivity (green and dry matter) based on Grassmaster II capacitance probe measurements. Considering the seasonal changes in pasture moisture content (PMC), it is clear that the more favorable period for the use of the Grassmaster II probe in dryland biodiverse pastures in Alentejo region of Southern Portugal coincides with the end of winter and beginning of spring (February–March; PMC > 80%: r = 0.959; p < 0.01; RMSE = 3191 kg ha$^{-1}$; $CV_{RMSE}$ = 23.6% for GM; and r = 0.953; p < 0.01; RMSE = 647 kg ha$^{-1}$; $CV_{RMSE}$ = 27.3% for DM). From late spring onward (May–June), the rapid drop in PMC adversely affects the measurements and the probe accuracy. The estimate of productivity is crucial for the management of the animal grazing, in terms of animal stocking in each field, of

animal rotation through the different fields, the calculation of the supplementary feed and forage needs for the animals. It can be concluded that the capacitance probe is an expedient tool to support the decision-making process in the management of dryland pasture and its respective dynamic grazing. The results of this work open up good perspectives for an approach in future studies that evaluate if pasture productivity estimation accuracy throughout the growing season could be further improved by combining spectral reflectance measurements (obtained by RS or proximal sensing) with capacitance probe measurements.

**Author Contributions:** J.S. (≈40%): Conceptualization; formal analysis; funding acquisition, investigation, methodology, supervision, writing. S.S. (≈15%): Conceptualization, investigation, review and editing. F.M. (≈15%): Formal analysis, investigation, methodology. F.C.-R. (≈15%): Formal analysis, investigation, methodology. J.M.d.S. (≈15%): Conceptualization, formal analysis, investigation, methodology. All authors have read and agreed to the published version of the manuscript.

**Funding:** This work was funded by National Funds through FCT (Foundation for Science and Technology) under the Project UIDB/05183/2020 and by the projects PDR2020−101-030693 and PDR2020−101-031244 ("Programa 1.0.1-Grupos Operacionais").

**Conflicts of Interest:** The authors declare no conflict of interest.

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
