# Peer review of "Estimation of Productivity in Dryland Mediterranean Pastures: Long-Term Field Tests to Calibration and Validation of the Grassmaster II Probe"

_agriengineering, doi:10.3390/agriengineering2020015_

Round 1

Reviewer 1 Report

There are many publications on the use of the capacitance pasture probe and a number were referenced in this study.  Many people have worked with calibrating the pasture probe for their specific research project. This is a quality research paper, but this MS has a fairly narrow window of application- to the location where the research was done and to a lesser extent, places with diverse vegetation in their pastures. They do not present any new technology or new approach to calibration and limited applicability.

there are only two suggested improvements for the MS:

Page 3 figure 1 the label for PVC handle might be more appropriately labeled PVC sensor.

Page 6 instead of metal rim, metal quadrat would be a better word choice. 

Author Response

“Estimation of productivity in dryland Mediterranean pastures:
long-term field tests to calibration and validation of the Grassmaster II
probe”

Manuscript ID –“agriengineering-758580”

___________________________________________________

Reviewer 1- Comments and Suggestions for Authors

General comment:

There are many publications on the use of the capacitance pasture probe and a number were referenced in this study.  Many people have worked with calibrating the pasture probe for their specific research project. This is a quality research paper, but this manuscript has a fairly narrow window of application- to the location where the research was done and to a lesser extent, places with diverse vegetation in their pastures. They do not present any new technology or new approach to calibration and limited applicability.

R- The authors would like to thank the reviewers' comments and suggestions.  The inclusion of 11 experimental fields, with different characteristics (types of pasture, predominant species, grazing, etc.) represents the variability of the Montado ecosystem and extends the applicability and interest of this technology to supporting the decision-making process in the  management of dryland pastures.

There are only two suggested improvements for the manuscript:

Comment 1:  Page 3 figure 1 the label for PVC handle might be more appropriately labeled PVC sensor.

R- Thank you. Reviewer's suggestion was incorporated into the new version of the article.

Comment 2: Page 6 instead of metal rim, metal quadrat would be a better word choice. 

R- Thank you. Reviewer's suggestion was incorporated into the new version of the article.

Reviewer 2 Report

The main goal of the paper is to test the performance of the Grassmaster sensor in estimating green biomass (GM) and dry biomass (DM). I have a hard time finding this work innovative. It is just a simple linear regression between biomass and Grassmaster measurements. The introduction is weak and needs work. The authors tried to justify the benefits of Grassmaster by incorporating the drawback of remote sensing. However, it is incomplete, misunderstood and needs significant work to make a case. The method is not sound, the number of calibration and validations samples are highly different which makes the results spurious.

The wrong method also affects the main conclusion of the paper that “the more favorable period for the use of this probe in dryland pastures in a Mediterranean climate, such as the Portuguese Alentejo coincides with the end of winter and beginning of spring (February-March), corresponding to PMC > 80%”. Authors didn't have any validation or even calibration data outside of (February-March), thus the conclusion is misleading.  The English of the paper needs to be improved.

Some other comments:

  • “An alternative to calculating vegetation indices is to use unmanned aerial vehicles (UAV)…”. This is a wrong statement. UAV is a platform carrying remote sensing sensors, as in satellites. NDVI is a vegetation index (VI) that can be calculated from these sensors onboard UAV or satellites. Thus, UAV is not an alternative approach to NDVI.
  • “In the last few years, many such sensors have been developed and marketed, such as “Crop Circle”, “Yara N-Sensor”, “GreenSeeker” or “OptRx”. Please cite the producers of these sensors.
  • The calibration phase is 11 years (2007-2018) and the validation is just one year (2019). The number of 1411 samples for calibration and 48 samples for validation leads to overoptimistic results.

  • Figure 3 needs aesthetic works.

  • How did you upscale the biomass collected in plots (0.25 m × 0.40 m in size) to kg ha−1?

  • Why there are two categories of PMC in table 3?

  • What is the point of regression models for multiple PMC classes when you don’t have validation data for the same classes?

  • Figure 5 is based on the correlation coefficients and RMSE from the calibration period. This analysis should be done using validation data, not calibration.

  • Add a one-to-one plot of estimated vs measured GM and DM.

Author Response

“Estimation of productivity in dryland Mediterranean pastures:
long-term field tests to calibration and validation of the Grassmaster II
probe”

Manuscript ID –“agriengineering-758580”

___________________________________________________

Reviewer 2- Comments and Suggestions for Authors

R- The authors would like to thank the reviewers' comments and suggestions, which have greatly improved the final version of the article.

General comments (GC):

GC1: The main goal of the paper is to test the performance of the Grassmaster sensor in estimating green biomass (GM) and dry biomass (DM). I have a hard time finding this work innovative. It is just a simple linear regression between biomass and Grassmaster measurements. The introduction is weak and needs work. The authors tried to justify the benefits of Grassmaster by incorporating the drawback of remote sensing. However, it is incomplete, misunderstood and needs significant work to make a case. The method is not sound, the number of calibration and validations samples are highly different which makes the results spurious.

R- The “Introduction” section was revised and complemented accordingly to the reviewer's suggestions. Also according to the reviewer's suggestions, more validation samples have now been used (216 instead of 48 from the original version).

GC2: The wrong method also affects the main conclusion of the paper that “the more favorable period for the use of this probe in dryland pastures in a Mediterranean climate, such as the Portuguese Alentejo coincides with the end of winter and beginning of spring (February-March), corresponding to PMC > 80%”. Authors didn't have any validation or even calibration data outside of (February-March), thus the conclusion is misleading.  The English of the paper needs to be improved.

R- With the new data, validation and calibration phases correspond to the same period (between February and June), when the Grassmaster probe can be used in Mediterranean conditions. The English of the paper was revised by a native.

Some other comments (OC):

OC1: “An alternative to calculating vegetation indices is to use unmanned aerial vehicles (UAV)…”. This is a wrong statement. UAV is a platform carrying remote sensing sensors, as in satellites. NDVI is a vegetation index (VI) that can be calculated from these sensors onboard UAV or satellites. Thus, UAV is not an alternative approach to NDVI.

R- Correct. The UAV are an alternative to satellite imagery and not an alternative to remote sensing, both are solutions of remote sensing. In order to avoid any confusion, the statement was rewritten.

OC2: “In the last few years, many such sensors have been developed and marketed, such as “Crop Circle”, “Yara N-Sensor”, “GreenSeeker” or “OptRx”. Please cite the producers of these sensors.

R- Reviewer's suggestion was incorporated into the new version of the article.

OC3: The calibration phase is 11 years (2007-2018) and the validation is just one year (2019). The number of 1411 samples for calibration and 48 samples for validation leads to overoptimistic results.

R- The validation phase has been extended from 48 to 216 samples and 4 to  8 experimental fields.

OC4:  Figure 3 needs aesthetic works.

R- The suggestion was accepted. A new Figure 3 was incorporated into the manuscript.

OC5: How did you upscale the biomass collected in plots (0.25 m × 0.40 m in size) to kg ha−1?

In each experimental field was collected a number de pasture samples representative of total area, whereby biomass measurements (g/m2) can be transformed to kg/ha because all obtained trials in space and time represent on an average base the pasture yield productivity managed by the farmer, allowed the scaling-up measurements from plot level to hectare level.

OC6: Why there are two categories of PMC in table 3?

R- The reviewer is right, the information is redundant so tables 3 and 4 were corrected accordingly.

OC7: What is the point of regression models for multiple PMC classes when you don’t have validation data for the same classes?

R- New validation data for multiple PMC classes were included .

OC8: Figure 5 is based on the correlation coefficients and RMSE from the calibration period. This analysis should be done using validation data, not calibration.

R- Figure 5 (new Figure 6) is only a graphic representation based on some parameters (correlation coefficients and CVRMSE) of Tables 4 and 5 that can help to explain the influence of PMC on the accuracy of Grassmaster II probe measurements and estimation models.  

OC9: Add a one-to-one plot of estimated vs measured GM and DM.

R- Plots of estimated vs measured pasture GM and DM for multiple PMC classes were included.
